# Utility of Cardiac Power Hemodynamic Measurements in the Evaluation and Risk Stratification of Cardiovascular Conditions

**DOI:** 10.3390/healthcare10122417

**Published:** 2022-11-30

**Authors:** Jonathan Farshadmand, Zachary Lowy, Ofek Hai, Roman Zeltser, Amgad N. Makaryus

**Affiliations:** 1Donald and Barbara Zucker School of Medicine at Hofstra/Northwell, Hofstra University, 500 Hofstra Blvd., Hempstead, NY 11549, USA; jfarshadmand1@pride.hofstra.edu (J.F.); zlowy1@pride.hofstra.edu (Z.L.); rzeltser@numc.edu (R.Z.); 2Department of Cardiology, Nassau University Medical Center, Hempstead, NY 11554, USA; ohai@numc.edu

**Keywords:** cardiac power, prognosis of cardiac power, cardiac power index, cardiac power output

## Abstract

Despite numerous advancements in prevention, diagnosis and treatment, cardiovascular disease has remained the leading cause of mortality globally for the past 20 years. Part of the explanation for this trend is persistent difficulty in determining the severity of cardiac conditions in order to allow for the deployment of prompt therapies. This review seeks to determine the prognostic importance of cardiac power (CP) measurements, including cardiac power output (CPO) and cardiac power index (CPI), in various cardiac pathologies. CP was evaluated across respective disease-state categories which include cardiogenic shock (CS), septic shock, transcatheter aortic valve replacement (TAVR), heart failure (HF), post-myocardial infarction (MI), critical cardiac illness (CCI) and an “other” category. Literature review was undertaken of articles discussing CP in various conditions and this review found utility and prognostic significance in the evaluation of TAVR patients with a significant correlation between one-year mortality and CPI; in HF patients showing CPI and CPO as valuable tools to assess cardiac function in the acute setting; and, additionally, CPO was found to be an essential tool in patients with CCI, as the literature showed that CPO was statistically correlated with mortality. Cardiac power and the derived measures obtained from this relatively easily obtained variable can allow for essential estimations of prognostic outcomes in cardiac patients.

## 1. Introduction

Despite numerous advancements in prevention, diagnosis, evaluation and treatment, cardiovascular disease is the leading cause of morbidity and mortality in the US and globally, and has been for the past 20 years [1]. Due to the high mortality surrounding cardiovascular diseases, there has been extensive research into determining the function and prognostic utility of hemodynamic parameters and biomarkers. Some of the more commonly used prognostic variables include left ventricular ejection fraction (EF), cardiac output (CO), cardiac index (CI), pulmonary capillary wedge pressure (PCWP), blood pressure (diastolic, systolic and/or mean) and brain-natriuretic peptide (BNP). There are several issues, however, with using each of these variables, and their prognostic power varies across cardiac conditions. Both EF [2] and CO/CI [3] are dependent on preload and afterload and may therefore inaccurately represent cardiac function, especially in critical illness such as cardiogenic shock [4]. PCWP captures ventricular compliance and volume status more than actual cardiac pumping ability and it may therefore fail to predict symptomatic improvement [5] or outcome [6] in various cardiac conditions. Similarly, BNP is a measure of ventricular volume overload as opposed to cardiac performance and may therefore be normal in conditions such as heart failure with preserved ejection fraction (HFpEF), CS, and restrictive cardiomyopathies [7,8], despite severe disease. By combining both pressure and blood flow, cardiac power (CP) gives a better picture of the overall pumping ability of the heart. It can also be measured noninvasively using echocardiography, a common procedure in the care of cardiac patients and bioimpedance [9,10]. Specifically, resting CP is easily measured without having to subject patients to additional invasive procedures, which could lead to further decompensation in their condition. CP has been shown to be a powerful predictor of mortality in cardiogenic shock [6,11,12,13,14], chronic heart failure [15,16] and sepsis [17,18], among other conditions. Despite showing prognostic utility across a wide range of conditions, however, it is not routinely used to guide treatment of cardiac or critically ill patients. Hence, the goal of this review is to evaluate the prognostic utility of CP across a broad range of medical conditions as documented in the medical literature.

## 2. Materials and Methods

### 2.1. Search Process

The PubMed, Embase, Web of Science and Cochrane databases were searched for the term “cardiac power” in the title, abstract, or keyword from database inception through 2 February 2021. Only articles in English were included. In Embase and Web of Science, the search was further refined to include only full text citations. Relevant citations were imported into Covidence^®^ software and underwent de-duplication.

### 2.2. Eligibility

The inclusion criteria for this review were: English language, published, full text articles that related resting CP (CP output (CPO) and/or CP index (CPI)) directly to mortality (cardiovascular or all cause). Animal studies, case reports, abstracts, review articles, commentaries on published studies and studies using pediatric populations (18 and under) were excluded. Studies that evaluate a surrogate or proxy of CP (circulatory power, exercise CP, peak CP) instead of resting CP were also excluded, as were studies that did not relate CP directly to mortality and those that used CP as an endpoint instead of an independent variable.

### 2.3. Study Screening and Data Collection

Potential studies were screened by 2 reviewers, independently at 2 levels: first a title and abstract screen and then a full text review. Studies were selected for full text review if they discussed both CP and mortality in the abstract. Discrepancies at both stages were discussed and resolved by consensus. Data points recorded from each study include title, authors, medical condition examined, inclusion/exclusion criteria, outcomes and length of follow-up, population size, age, sex and baseline inotrope/vasopressor use, method of measuring and calculating CP, mean CP (overall and by group), CP cutoff and survival above and below the cutoff if applicable, summary statistics relating CP to the endpoint (relative risk (RR), odds ratio (OR), hazard ratio (HR), Wald/chi square, area under the receiver operating characteristic curve (AUC), c-statistic, confidence interval (CI)) and the variables adjusted for in the multivariate analysis if one was done. If multiple summary statistics were reported in a single study, all were recorded.

The prognostic utility of CP was evaluated across all included studies. It was further evaluated according to the medical condition being examined. Each article was analyzed by both reviewers and designated a specific category. Based on these designations, categories were created that grouped similar articles shown in Table 1, which indicates the categorical name as well as which articles were included within each group. Of note, a category named “other” was created based on individual articles that did not fit into a larger designated category. The “other” category included post extracorporeal circulation, non-cardiac critical illness, ischemic cardiomyopathy, heart transplantation and chronic kidney disease.

### 2.4. Bias Screening

The quality of bias within each article was assessed by both screeners using the Quality in Prognostic Studies (QUIPS) tool. Within the bias screen, each article was screened for study participation, study attrition, prognostic factor measurement, outcome measurement, study confounding and statistical analysis and reporting, individually. Each section was designated a High, Moderate, or Low bias [50]. The high bias in each section was given a score of 2, moderate was given a 1 and low was given a 0. Based on these six categories of biases, each article was designated an overall bias score of low, mild, moderate, or high by combining the scores of their individual 6 categories. An article was designated of an overall high bias if the total score of individual categories summed between 9–12; a score of moderate if the summation was between 6–8; mild if the score was between 3–5; and low bias if the summation was 0–2. Overall, out of the 41 articles, a total of 36 articles were determined to have low bias, 5 total articles had mild bias and zero articles were designated to have moderate or high bias. Individual article overall and subcategory bias designation can be seen in Table 2.

## 3. Results

A total of 1125 potential studies were imported for screening into Covidence^®^ (350 from PubMed, 378 from Embase, 334 from Web of Science and 63 from Cochrane), from which 1084 were excluded: 639 duplicates were removed, 353 were deemed irrelevant during the title and abstract stage and 92 during the full text screen (24 used CP as an endpoint instead of an independent variable, 21 evaluated a surrogate or proxy of CP instead of resting CP, 18 did not relate resting CP to mortality and/or morbidity, 10 were review articles and did not present any new data, 8 were duplicates, 5 were abstracts, 4 were unable to be retrieved, 1 was a commentary on a study and 1 did not have sufficient power to draw significant conclusions from the data). A schematic of the screening process is shown below in Figure 1. Detailed journal article demographics, outcomes, and disease state definitions of the condition examined are highlighted in Table 3. In the final analysis, 41 studies [3,6,11,12,13,14,15,16,17,18,19,20,21,22,23,24,25,26,27,28,29,30,31,32,33,34,35,36,37,38,39,40,41,42,43,44,45,46,47,48,49] were included covering a total of 33,906 patients. When determining the use of CP in the determination of prognosis throughout the various disease processes, we categorized the articles based on the disease process they were associated with, as shown in Table 1.

### 3.1. Aortic Stenosis

With regards to transcatheter aortic valve replacement (TAVR) due to symptomatic severe aortic stenosis (AS), it was found that all four studies in this category showed a strong and significant correlation between CPI and mortality, where the lowest study population was over 750 subjects. All of these articles calculated and compared CPI post-TAVR placement between living and deceased groups, where one article found a *p* < 0.03 [22] whereas the other three articles showed a significance of *p* < 0.001 [19,20,21]. Two of the articles were also able to determine very similar CPI cutoff values (0.49 [21] and 0.48 [19]) that were found to be statistically significant in determining one-year mortality. Lastly, one article showed that CPI had the highest relative importance in predicting 1-year mortality post TAVR placement, being approximately three times more impactful than the second most influential variable [20].

### 3.2. Septic Shock

This group contained three articles, where all three articles compared CPI between survivors and non-survivors and had slightly different definitions of septic shock, shown in Table 3. One study showed a very strong significance (*p* < 0.0001) with a population size of 39 [23]; another showed just under statistical significance between groups with *p* = 0.058 with a population size of 70 [17]; and the third article, which had a population size of 141, showed no statistical significance with *p* > 0.05 [18]. One article also tested the significance of CPO in determining mortality in septic shock and compared ICU survivors and non-survivors with a significant finding of *p* < 0.0001 [23]. However, of the studies that performed univariate analysis, they all compared CPI to one-month mortality approximately and both showed a significance of *p* < 0.05 [17,18]. One of the studies also performed a multivariate study which showed that 28-day mortality was independently associated with a decrease in CPI with OR: 1.84 [1.18–2.87] per 0.1 W/m^2^ decrease in CPI; *p* = 0.008 [17].

### 3.3. Post-Myocardial Infarction

The post-MI group only had two articles in this category, as patients who had an MI, which was determined based on ST-segment elevation that rapidly resulted in heart failure or cardiogenic shock, were placed in the HF or cardiogenic shock category, respectively. Both of these studies measured in-hospital mortality using thoracic electrical bioimpedance to measure CPI and CPO [24,25], but one study continued to follow their patients, also measuring 1- and 5-year mortality [24]. The study that only measured in-hospital mortality had a sample size of 232 [25]. This study initially showed CPO and CPI to be significantly correlated with diabetes (*p* < 0.001 for CPI, *p* = 0.005 for CPO), which then went on to show that, for all patients, diabetic and non-diabetic, CPO and CPI were significant in univariate analysis with in-hospital mortality within all three groups, but not on multivariate analysis [25]. The second article, which had a patient population of 208, showed that CPO and CPI was determined to be significant at in-hospital mortality, 1-year mortality and 5-year mortality in univariate analysis. (CPO at 5-year mortality had a HR of 4.33 [1.73–10.84] with *p* = 0.002). However, on multivariate analysis CPI was not significant with in-hospital mortality, 1-year mortality, or 5-year mortality. When determining the significance on multivariate analysis for CPO, it was found to only be significant for 5-year mortality [24].

### 3.4. Heart Failure

Within the heart failure group, the definition or context around heart failure varied by a significant amount based on each article. For example, some articles were in the context of an acute exacerbation while others were in a stable chronic heart failure setting. Specifics for each article can be found in Table 3. One of the largest groups was heart failure (HF), which we further divided into articles that looked at CPI and articles that looked at CPO, as there were no articles that looked at both measurements. For CPI, there was a total of two articles, with one article incorporating 28 patients [29], while the other had a total of 495 [16]. The smaller study compared CPI to mortality at 90 days and found no statistical difference in resting CPI between survivors and non-survivors, but did not perform univariate or multivariate analysis [29]. The larger study’s endpoints were all cause mortality, heart transplant, or ventricular assistant device placement and had an average follow up of 3.3 years [16]. This article created a CPI cut-off of 0.44 to determine mortality between these two groups and this had a Chi-square of 43.9, *p* < 0.0001. Through univariate analysis, CPI remained significant showing a 2.4-fold increase in mortality in the lower CPI group (HR 2.38, 95%CI 1.83–3.11, *p* < 0.0001). Multivariate analysis with adjusted CPI also remained independently associated with the study’s endpoints (HR 1.52, 95% CI 1.03–2.28, *p* = 0.04) [16].

With regards to the HF group when looking into CPO, there were a total of 10 articles with a mean, median and range of the population sample of 20,478; 160; and 50–18,733, respectively. The study duration and endpoints varied between articles, but the majority of the articles looked at mortality for at least one year. Of the ten articles, 7 articles compared resting CPO to their correlated endpoint, where 5/7 articles showed strong significance between CPO and prognosis. Two articles also produced CPO cut-off values to determine resting CPO’s correlation with prognosis. One of the articles created a CPO cutoff of 0.54 which showed that patients below a CPO of 0.54 have a significantly lower probability of survival (*p* < 0.0001) and should be classified as high-risk patients [36]. The other study created a CPO cutoff value of 0.6, which showed that falling below this value increased the odds of worsening heart failure by 4.0 and 2.6 at 7 days and 30 days, respectively. However, at the 6-month mark, baseline CPO < 0.6 was no longer found to be significant with an OR of 0.4. The article also found that a decrease in CPO at the 6 hr mark showed OR of 3.4, 3.0 and 2.0 at the 7-day, 30 day and 6 months follow up marks. Lastly, patients who met an initial baseline CPO < 0.6 and decrease in CPO at the 6 h mark, were found to have OR = 8.3, 95% CI = 1.2–56.0, *p* = 0.004 at 7 days, OR = 14.3, 95% CI = 2.7–76.5, *p* = 0.002 at 30 days and OR = 3.1, 95% CI = 0.6–15.5, *p* = 0.174 at 6 months [34]. Upon univariate analysis, a total of 8 articles examined the relationship between CPO and prognosis, where only 3 [15,34,36] showed statistical significance. Furthermore, upon multivariate analysis, only 1 [36] out of 7 articles found any significant relationship between CPO and prognosis.

### 3.5. Cardiogenic Shock

The cardiogenic shock category was the largest category with 13 articles included, so, similarly to the HF group, this group was also divided into articles that discussed CPI or CPO. Within these articles, the definition of cardiogenic shock had some slight variation, but for the majority was defined by SBP < 90 mmHg for >30 min or use of inotropes/vasopressors to maintain a SBP > 90 mmHg, some evidence of end-organ hypoperfusion and caused by a cardiogenic etiology. However, in this group, there were two articles that discussed CPI and CPO together, so their respective findings were included into each category. There was a total of 10 articles with regards to CPI in cardiogenic shock, with most using mortality as an endpoint but at various follow-up intervals. The articles had a mean, median and range of population sample of 163, 91 and 68–541, respectively. Of the ten articles, four did not compare CPI to an endpoint or another group and only measured the median CPI of the patients. The other 6 articles which compared CPI between survivors and non-survivors included 5/6 showing statistical significance [13,37,40,41,44]. Of the ten articles, 7 performed univariate and multivariate analysis which all found CPI to be statistically significant in both analyses [6,12,13,14,39,40,44]. Only four articles performed any analysis with regards to prognosis with the use of a CPI cut-off value. One article showed an AUC = 0.81 with sensitivity = 39.3 and specificity = 88.9, when hourly time integral of CPI drops below 0.8. However, when using a cut-off of 0.4, the analysis showed an AUC = 0.79, Sensitivity = 39.3 and Specificity = 90.5 [44]. Another study found that CPI < 0.2 had 80% specificity for 28-day mortality [40]. The last study combined having CPI < 0.28 with a history of ischemic cardiomyopathy and this showed that chance for survival was less than 20% in 4 weeks, *p* = 0.005 [13]. These three articles all showed varying degrees of using cut-off values for CPI in determining prognosis, but strong specificity in helping rule in patients with poor prognosis in cardiogenic shock.

There was a total of five articles that discussed CPO in cardiogenic shock, including the two articles that discussed both CPO and CPI simultaneously. Of the five articles, 4/5 measured mortality as their endpoint at various time points and the mean, median and range of the population samples was 204, 171 and 28–541, respectively. Of the five articles, three discussed mean CPO with regards to prognosis, whereas the other two only reported their group’s average CPO, but did not determine significance or compare it to any alternate group. Of the three articles, one article found CPO not to be statistically significant *p* = 0.0625; another article also found it not to be significant, but found *p* > 0.01 and did not include the actual *p*-value [43], and the last article found CPO to be significant at two time points after their procedure, but found that pre-operative CPO was not significant between survivors and non-survivors [38]. Within this group, four of the articles created CPO cut offs, where one article found that all patients with CPO < 0.35 died in 1 year follow up [43]. The three other articles created similar cut-off values [6]. One of the articles with a cut-off of 0.6 showed that it was significantly correlated to prognosis [11], whereas the last one calculated a sensitivity of 38% and specificity of 88% in predicting mortality [38]. Only three of the articles performed univariate and multivariate analysis, but all three showed the significance of CPO with mortality [6,11,38].

### 3.6. Critical Cardiac Disease

Another category was critical cardiac disease in the CCU or CICU which included two studies. The definition of critical cardiac disease varied between the two studies. As for the CCU group, it was defined as having a primary cardiac diagnosis undergoing PAC, where for the CICU group it was defined as patients admitted to the CICU and meeting any stage of the Society for Cardiovascular Angiography and Intervention for shock. Both of these studies’ end points were in-hospital mortality and compared this to CPO. The CICU study had a total of 5453 patients [26] while the CCU study had a total of 349 patients [3]. The smaller study found that, when using a CPO cut-off of 0.53, 49% of patients with CPO ≤ 0.53 died in the hospital, while only 20% with CPO > 0.53 died in the hospital and this was shown to be significant with a *p* < 0.001, PPV of 49% and NPV of 80% [3]. Furthermore, upon univariate analysis, CPO was found to have OR of 0.66, 95% CI of 0.55–0.79 and *p* < 0.001. On multivariate analysis of CPO, it showed an OR of 0.65, 95% CI of 0.54–0.78, *p* < 0.001 [3]. In the CICU study, the patients were stratified into stages based on the Society for Cardiovascular Angiography and Interventions. Within each stage, the median CPO and CPI were calculated as well as the percentage of patients that had a CPO < 0.6. The CPI, CPO and CPO < 0.6 groups per stage all showed great significance with *p* < 0.001. Within multivariate analysis of CPO, the adjusted OR for having a CPO < 0.6 was 1.859 with 95% CI 1.291–2.678 and *p* < 0.001, without including other variables in the analysis. The adjusted OR for CPO < 0.6 when combined with other variables, dropped to 1.504 with 95% CI of 0.939–2.408, *p* = 0.09 and was no longer statistically significant [26].

### 3.7. Other Diseases

A few articles included were not able to be fit into a categorical section with other similar papers, which is why an “Other” category was created.

#### 3.7.1. Non-Cardiac Related Illness

One article in this group determined the use of CPO and CPI in non-cardiac related illness based on their correlation to in-hospital mortality for 32 patients. Initial hospital admission CPO for survivors had a median CPO of 1.14 (IQR: 0.9–1.52) and non-survivors had a median CPO of 0.89 (0.67–0.99) which had a *p* < 0.05. Within this study, survivors and non-survivors were also compared at optimal (t_1_) and max (t_max_) fluid resuscitation for CPO and this showed a significant difference between the two groups at these additional time points; t_1_ *p* < 0.0001 and t_max_ *p* < 0.0001. CPI for the survivors was found to be 0.48 and for non-survivors was 0.51, which was found not to be significant with *p* > 0.05; however, at t_1,_ CPI for survivors was 0.88, while for non-survivors this was 0.57 with *p* < 0.05 [49].

#### 3.7.2. Heart Transplant

An article on heart transplant compared CPI for 140 patients using primary graft dysfunction (PGD) in 30 days as their primary end point. Patients with no severe PGD had an initial median CPI of 0.44 (IQR: 0.37–0.53) while patients with severe PGD had median CPI of 0.29 (IQR: 0.23–0.33) with *p* < 0.001. Patients were also seen to have significant CPI differences at the 6-h mark, *p* < 0.001. Furthermore, the study was able to show that using an initial CPI cut off of 0.34 and a CPI cutoff of 0.33 at the six-hour mark showed that 79% of patients who had CPI values above cut-off values at their respective time points, survived, while patients who were below both cut-off points only had a survival rate of 2%, showing a high NPV [48].

#### 3.7.3. Chronic Kidney Disease

Another article dealt with CPI in chronic kidney disease with 349 patients and found that patients with a low CPI defined as MAP <  88 mmHg and CI  <  3.25  L/min/m^2^ had the highest one year mortality rate of 23.4%, whereas patients with a high CPI, defined as MAP ≥ 88 mmHg and CI  ≥  3.25  L/min/m^2^, had a mortality rate of 5.6% with *p* = 0.06 and with no univariate or multivariate analysis carried out in the study on CPI [47].

#### 3.7.4. Ischemic Cardiomyopathy

The next article was in regard to ischemic cardiomyopathy which mainly focused on peak CPO and included 111 patients, but did find significance in univariate analysis of resting CPO in these patients. Resting CPO had a HR of 1.10 (95% CI: 0.33; 3.64) with regards to mortality, with median follow up of 29 months [46].

#### 3.7.5. Extracorporeal Circulation

Lastly, Clark et al.’s article with regards to post extracorporeal circulation for cardiac procedures analyzed CPO and CPI in relation to survival 7 days post-procedure in 181 patients. This article showed that CPO and CPI of the deceased patients in the study was approximately half the value of surviving patients with *p* < 0.03, but they were unable to perform univariate or multivariate analysis due to the small number of deaths in the patient pool [45].

## 4. Discussion

The prognostic implications of cardiac power were analyzed across 41 articles and this evaluation showed prognostic use across various disease processes with varying degrees of utility. CPI within TAVR patients was noted to be a profound marker in one year mortality as all articles showed statistical significance. Furthermore, these articles showed the importance of creating a standardized CPI cut-off value to identify high risk patients after TAVR placement as the articles cut off values were 0.48/0.49.

Overall, within the septic shock category, there was much more variation between study results. Out of the three articles, one showed strong significance, one showed no significant correlation and one was just on the border of significance. However, upon univariate and multivariate analysis, CPI was found to be significantly correlated to patient mortality. Furthermore, CPO was also found to be significantly correlated to mortality in septic shock, but was only looked at in one article, making it more difficult to produce a significant takeaway. Overall, the mixed results indicate potential for the use of CPI and CPO in septic shock, but would likely need further analysis of CP within septic shock for more definitive determination of its use.

Even though CPO and CPI were found to be correlated to mortality in post-MI patients on the univariate analysis in both articles, it was no longer significant on multivariate analysis at all time points tested for CPI and was only significant for CPO at the five-year mark. Overall, this indicates that CPI would not be a good prognostic indicator for post-MI patients and most likely neither is CPO, but CPO shows some potential especially in long term prognostic evaluation and needs further research.

Overall, the use of CPI in HF was split between two articles, as half showed no association of CPI with mortality, while the other showed strong statistical significance. However, on further analysis, including QUIPS screening, the two articles were not shown to be equivalent with regards to the study’s rigor and bias screening. The article that indicated that there was no significant relationship between CPI and prognosis in HF incorporated only 28 patients and followed patients for 90 days and this may have led to a larger degree of variability. Additionally, this article on QUIPS bias screen was found to be one of the five articles with a mild bias screen, indicating an overall less reliability. However, the other article which contained 495 patients had a much longer follow-up time (3.3 years) and QUIPS bias screening of low was able to show strong significant correlation between CPI and mortality upon univariate, multivariate and using a CPI cut off of 0.44.

The use of CPO in HF was significantly analyzed as it was the second largest group and it showed CPO to be significantly related to mortality when comparing groups on univariate analysis, but fell short in the majority of articles on multivariate analysis, making CPO an unreliable prognostic tool in HF, unlike CPI. However, using a standardized CPO cut-off (0.60 and 0.54) value saw potential in helping identify high risk patients across multiple articles in the acute setting; a CPO cutoff alone should not be used in a long-term setting, but instead could perhaps be used in combination with other cardiac power markers or CPO values at different time points.

The use of CPI in cardiogenic shock patients was also widely analyzed and showed that CPI is a very strong predictor of prognosis in cardiogenic shock. Furthermore, even though the cut-off values of CPI were not as similar as those seen in other pathological categories, they showed a strong trend towards prognosis, indicating the use of CPI as a cut-off to determine patients with high mortality, and showed a much stronger ability to rule out patients from having low prognostic status in cardiogenic shock. rather than in ruling them in. CPI was certainly found to be a good indicator of mortality in cardiogenic shock patients, but in regard to what cut-off value should be used to stratify low and high-risk patients, further studies and analysis would need to be performed to create a standardized cut-off value.

CPO in cardiogenic shock did not show as strong and uniform results as CPI. Few articles found no correlation, whereas the others found CPO to be a good indicator of prognosis. Due to these mixed results, the validity of CPO in cardiogenic shock was conservatively indicated not to be useful, but would certainly require a meta-analysis to interpret the data more rigorously. Nevertheless, CPO did show some use, especially in creating a cut-off value, as both studies noted similar CPO cut-off values (0.60 and 0.53) which were also similar to those noted for HF patients. This shows potential as it can allow providers to identify high risk patients under these cut-off values and possibly change treatment plans to decrease overall mortality for these patient populations.

CPO in critical cardiac disease patients showed utility in determining mortality. Both studies which contain over hundreds of patients showed CPO to be significant in all analyses performed. Furthermore, both studies created CPO cut off values that were also very similar, 0.60 and 0.53, and both showed strong significance, indicating that stratifying patients based on these cut-offs, and possibly altering treatment or monitoring, would be very useful. Lastly, CPO < 0.6 maintained significance, but when incorporating other variables on multivariate analysis, the adjusted value no longer had statistical significance thus indicating the importance of CPO over other variables.

Determining the overall utility of CP within the “other” category proved to be slightly more challenging as the articles were limited in number and scope. For non-cardiac related illness, only CPO was found to be prognostic with regards to mortality. For heart transplantation, CPI was found to be very useful in determining PGD and was also able to create CPI cut-offs at various time points in order to predict graft dysfunction. For patients with CKD, CPI was combined with other variables and not tested in isolation, but even with the addition of other variables, was still not found to be significant, indicating that it would not provide much clinical use, especially in isolation for these patients. The article on ischemic cardiomyopathy only briefly analyzed resting CPO, but found it to be significant in univariate analysis. Lastly, in patients with post extracorporeal circulation for cardiac procedures, CPO and CPI showed promising results but was limited. All articles found within this group should be taken in the context that the articles were found in isolation, indicating that all of these topics need further research with regards to their utility in a clinical setting.

### Study Limitations

There are several potential limitations to the current review. First, grey literature abstracts and non-English publications were excluded, which could have changed the proportion of studies that found CP to be prognostically significant and introduced publication bias. Most of the studies included are observational studies and the review may be subject to bias inherent to observational study designs. Due to the heterogeneity present in the literature evaluating cardiac power, comparing the effect sizes across studies was not feasible, limiting the comparative interpretations that can be drawn. For example, formulation calculation or measurement technique/procedure was not uniform throughout all articles and this may result in bias. However, the goal of the current study was to give a broad overview of the state of the literature regarding resting CP and more general patterns in the data were able to be compared. Furthermore, when breaking up the articles by pathological process, the method of measurement and formula for calculating CP was much more similar, but not exactly the same in certain disease processes. Additionally, the criteria or definition for each disease process such as cardiogenic shock or heart failure was not identical within each group, leading to further discrepancies between studies, which does make establishing a uniform conclusion more difficult. In the future, conducting studies that use a more homogenized approach for measuring, recording and analyzing CP data would be informative and may allow for more rigorous evaluation of CP across a range of cardiac conditions to allow for a more robust analysis on the efficacy of CP as a prognostic tool across various diseases.

## 5. Conclusions

Cardiac power is documented in the literature as an impactful tool in identifying patients with various cardiac and critical conditions at higher risk of mortality. Specifically, CP was found to be useful in patients who received TAVR (CPI), who have HF (CPI), cardiogenic shock (CPI) and critical cardiac illness (CPO). Lastly, there is a strong demand for and utility in creating standardized CPO and CPI cut-off values for the following conditions: patients post-TAVR (CPI), HF (CPI), cardiogenic shock (CPI and CPO) and critical cardiac illness (CPO). Further study of this clinically useful and relatively easy to obtain parameter of cardiac power should be performed.

**Table 2 healthcare-10-02417-t002:** QUIPS bias screening results [3,6,11,12,13,14,15,16,17,18,19,20,21,22,23,24,25,26,27,28,29,30,31,32,33,34,35,36,37,38,39,40,41,42,43,44,45,46,47,48,49,50].

Article Title	Authors	Year	Journal	Study Participation Bias	Study Attrition Bias	Prognostic Factor Measurement Bias	Outcome Measurement Bias	Study Confounding Bias	Statistical Analysis and Reporting	Overall Bias Score
Prognosis of in-hospital myocardial infarction course for diabetic and nondiabetic patients using a noninvasive evaluation of hemodynamics and heart rate variability.	Ablonskytė-Dūdonienė, R.; Bakšytė, G.; Ceponienė, I.; Kriščiukaitis, A.; Drėgūnas, K.; Ereminienė, E.	2013	Medicina (Kaunas)	Low	Low	Low	Low	Low	Low	Low
Impedance cardiography and heart rate variability for long-term cardiovascular outcome prediction after myocardial infarction.	Ablonskytė-Dūdonienė, R.; Bakšytė, G.; Čeponienė, I.; Kriščiukaitis, A.; Drėgūnas, K.; Ereminienė, E.	2012	Medicina (Kaunas)	Low	High	Low	Low	Low	Low	Low
Does Resting Cardiac Power Index Affect Survival Post Transcatheter Aortic Valve Replacement?	Agasthi, P.; Arsanjani, R.; Mookadam, F.; Wang, P.; Venepally, N.R.; Sweeney, J.; Eleid, M.; Holmes, D.R., Jr; Pollak, P.; Fortuin, F.D.	2020	Journal Invasive Cardiology	Low	Moderate	Low	Low	Low	Low	Low
Artificial intelligence trumps TAVI(2)-SCORE and CoreValve Score in predicting 1-year mortality post Transcatheter Aortic Valve Replacement.	Agasthi, P.; Ashraf, H.; Pujari, S.H.; Girardo, M.E.; Tseng, A.; Mookadam, F.; Venepally, N.R.; Buras, M.; Khetarpal, B.K.; Allam, M.; Eleid, M.F.; Greason, K.L.; Beohar, N.; Siegel, R.J.; Sweeney, J.; Fortuin, F.D.; Holmes, D.R., Jr; Arsanjani, R.	2020	Cardiovascular Revascularization Medicine	Low	Moderate	Low	LOW	Low	Low	Low
Does a Gradient-Adjusted Cardiac Power Index Improve Prediction of Post-Transcatheter Aortic Valve Replacement Survival Over Cardiac Power Index?	Agasthi, P.; Pujari, S.H.; Mookadam, F.; Tseng, A.; Venepally, N.R.; Wang, P.; Allam, M.; Sweeney, J.; Eleid, M.; Fortuin, F.D.; Holmes, D.R.; Beohar, N.; Arsanjani, R.	2020	Yonsei Medical Journal	Low	moderate	Low	LOW	Low	Low	Low
Differential responses to larger volume intra-aortic balloon counter-pulsation: Hemodynamic and clinical outcomes.	Baran, D.A.; Visveswaran, G.K.; Seliem, A.; DiVita, M.; Wasty, N.; Cohen, M.	2018	Catheterization and Cardiovascular Interventions	Moderate	Moderate	Low	Low	Low	Low	Low
Improved Outcomes Associated with the use of Shock Protocols: Updates from the National Cardiogenic Shock Initiative.	Basir, M.B.	2019	Catheter Cardiovasc Interv	Low	Low	Low	Low	Low	Low	Low
The value of cardiac output studies in postoperative cardiac patients: a myth	Clark, R.E.; Siegfried, B.A.; Ferguson, T.B.	1976	Journal of Surgical Research	Moderate	Moderate	Moderate	Low	Moderate	Low	Mild
Prognostic value of cardiac power output to left ventricular mass in patients with left ventricular dysfunction and dobutamine stress echo negative by wall motion criteria.	Cortigiani, L.; Sorbo, S.; Miccoli, M.; Scali, M.C.; Simioniuc, A.; Morrone, D.; Bovenzi, F.; Marzilli, M.; Dini, F.L.	2017	European Heart Journal—Cardiovascular Imaging	Low	Low	Low	Low	Low	Low	Low
Impaired microcirculation predicts poor outcome of patients with acute myocardial infarction complicated by cardiogenic shock.	den Uil, C.A.; Lagrand, W.K.; van der Ent, M.; Jewbali, L.S.; Cheng, J.M.; Spronk, P.E.; Simoons, M.L.	2010	European Heart Journal	Low	Low	Low	Low	Low	Low	Low
Non-invasive hemodynamic profiling of patients undergoing hemodialysis—A multicenter observational cohort study.	Doenyas-Barak, K.; de Abreu, M.H.F.G.; Borges, L.E.; Tavares Filho, H.A.; Yunlin, F.; Yurong, Z.; Levin, N.W.; Kaufman, A.M.; Efrati, S.; Pereg, D.; Litovchik, I.; Fuchs, S.; Minha, S.	2019	BMC Nephrology	Low	Moderate	Low	Low	Low	Low	Low
Cardiac power is the strongest hemodynamic correlate of mortality in cardiogenic shock: a report from the SHOCK trial registry.	Fincke, R.; Hochman, J.S.; Lowe, A.M.; Menon, V.; Slater, J.N.; Webb, J.G.; LeJemtel, T.H.; Cotter, G.	2004	Journal of the American College of Cardiology	Low	Moderate	Low	Low	Low	Low	Low
Emergency transcatheter aortic valve replacement in patients with cardiogenic shock due to acutely decompensated aortic stenosis.	Frerker, C.; Schewel, J.; Schlüter, M.; Schewel, D.; Ramadan, H.; Schmidt, T.; Thielsen, T.; Kreidel, F.; Schlingloff, F.; Bader, R.; Wohlmuth, P.; Schäfer, U.; Kuck, K.H.	2016	Eurointervention	Low	Moderate	Low	Low	High	Low	Mild
Experience with the Impella recovery axial-flow system for acute heart failure at three cardiothoracic centers in Sweden.	Granfeldt, H.; Hellgren, L.; Dellgren, G.; Myrdal, G.; Wassberg, E.; Kjellman, U.; Ahn, H.	2009	Scandinavian Cardiovascular Journal	Low	Low	Low	Low	Moderate	Low	Low
Prognostic role of cardiac power index in ambulatory patients with advanced heart failure.	Grodin, J.L.; Mullens, W.; Dupont, M.; Wu, Y.; Taylor, D.O.; Starling, R.C.; Tang, W.H.	2015	European Journal of Heart Failure	Low	Low	Low	Low	Moderate	Low	Low
Right Atrial Pressure Predicts Mortality Among LVAD Recipients: Analysis of the INTERMACS Database.	Guglin, M.; Omar, H.R.	2020	Heart, Lung and Circulation	Moderate	Low	Low	Low	Moderate	Low	
Cardiac power index: staging heart failure for mechanical circulatory support.	Hall, S.G.; Garcia, J.; Larson, D.F.; Smith, R.	2012	Perfusion	Moderate	Low	Low	Low	High	Low	
Predictors of intra-aortic balloon pump hemodynamic failure in non-acute myocardial infarction cardiogenic shock.	Hsu, S.; Kambhampati, S.; Sciortino, C.M.; Russell, S.D.; Schulman, S.P.	2018	American Heart Journal	Low	Moderate	Moderate	Low	Moderate	Low	
Hemodynamic parameters are prognostically important in cardiogenic shock but similar following early revascularization or initial medical stabilization—A report from the SHOCK trial	Jeger, R.V.; Lowe, A.M.; Buller, C.E.; Pfisterer, M.E.; Dzavik, V.; Webb, J.G.; Hochman, J.S.; Jorde, U.P.; Shock Investigator	2007	Chest	Low	High	Low	Low	Low	Low	
Noninvasive Hemodynamic Assessment of Shock Severity and Mortality Risk Prediction in the Cardiac Intensive Care Unit	Jentzer, J.C.; Wiley, B.M.; Anavekar, N.S.; Pislaru, S.V.; Mankad, S.V.; Bennett, C.E.; Barsness, G.W.; Hollenberg, S.M.; Holmes, D.R.; Oh, J.K.	2021	JACC: Cardiovascular Imaging	Low	Moderate	Low	Low	Low	Low	
Cardiac contractile reserve parameters are related to prognosis in septic shock.	Kimmoun, A.; Ducrocq, N.; Mory, S.; Delfosse, R.; Muller, L.; Perez, P.; Fay, R.; Levy, B.	2013	BioMed Research International	Low	Low	Low	Low	Moderate	Low	
Cardiac Power Output Index and Severe Primary Graft Dysfunction After Heart Transplantation.	Lim, H.S.; Ranasinghe, A.; Chue, C.; Quinn, D.; Mukadam, M.; Mascaro, J.	2021	Journal of Cardiothoracic and Vascular Anesthesia	Low	Low	Low	Low	Low	Low	
Circulating angiopoietins and cardiovascular mortality in cardiogenic shock.	Link, A.; Pöss, J.; Rbah, R.; Barth, C.; Feth, L.; Selejan, S.; Böhm, M.	2013	European Heart Journal	Low	Low	Low	Low	Moderate	Low	
Cardiac power output predicts mortality across a broad spectrum of patients with acute cardiac disease.	Mendoza, D.D.; Cooper, H.A.; Panza, J.A.	2007	American Heart Journal	Low	Low	Low	Low	Moderate	Low	
Cardiopulmonary and Noninvasive Hemodynamic Responses to Exercise Predict Outcomes in Heart Failure	Myers, J.; Wong, M.; Adhikarla, C.; Boga, M.; Challa, S.; Abella, J.; Ashley, E.A.	2013	Journal of Cardiac Failure	Low	Low	Low	Low	Moderate	Low	
Cardiac power index, mean arterial pressure and Simplified Acute Physiology Score II are strong predictors of survival and response to revascularization in cardiogenic shock.	Popovic, B.; Fay, R.; Cravoisy-Popovic, A.; Levy, B.	2014	Shock	Low	Low	Moderate	Low	Moderate	Low	
Echo-derived peak cardiac power output-to-left ventricular mass with cardiopulmonary exercise testing predicts outcome in patients with heart failure and depressed systolic function.	Pugliese, N.R.; Fabiani, I.; Mandoli, G.E.; Guarini, G.; Galeotti, G.G.; Miccoli, M.; Lombardo, A.; Simioniuc, A.; Bigalli, G.; Pedrinelli, R.; Dini, F.L.	2019	European Heart Journal Cardiovascular Imaging	Low	Low	Low	Low	Low	Low	
The short-term prognosis of cardiogenic shock can be determined using hemodynamic variables: a retrospective cohort study*.	Rigamonti, F.; Graf, G.; Merlani, P.; Bendjelid, K.	2013	Critical Care Medicine	Low	Moderate	Low	Low	Moderate	Low	
Peak cardiac power measured noninvasively with a bioreactance technique is a predictor of adverse outcomes in patients with advanced heart failure.	Rosenblum, H.; Helmke, S.; Williams, P.; Teruya, S.; Jones, M.; Burkhoff, D.; Mancini, D.; Maurer, M.S.	2010	Congestive Heart Failure	Low	Moderate	Low	Low	Moderate	Low	
Prognostic factors of chronic heart failure in NYHA class II or III: value of invasive exercise hemodynamic data.	Roul, G.; Moulichon, M.E.; Bareiss, P.; Gries, P.; Koegler, A.; Sacrez, J.; Germain, P.; Mossard, J.M.; Sacrez, A.	1995	European Heart Journal	Low	Low	Low	Low	Low	Low	
Current Use and Impact on 30-Day Mortality of Pulmonary Artery Catheter in Cardiogenic Shock Patients: Results From the CardShock Study.	Sionis, A.; Rivas-Lasarte, M.; Mebazaa, A.; Tarvasmäki, T.; Sans-Roselló, J.; Tolppanen, H.; Varpula, M.; Jurkko, R.; Banaszewski, M.; Silva-Cardoso, J.; Carubelli, V.; Lindholm, M.G.; Parissis, J.; Spinar, J.; Lassus, J.; Harjola, V.P.; Masip, J.	2020	Journal of Intensive Care Medicine	Low	High	Low	Low	Moderate	Low	
Measurement of cardiac reserve in cardiogenic shock: implications for prognosis and management.	Tan, L.B.; Littler, W.A.	1990	British Heart Journal	Low	Low	Low	Low	Low	Low	
Standardized Team-Based Care for Cardiogenic Shock.	Tehrani, B.N.; Truesdell, A.G.; Sherwood, M.W.; Desai, S.; Tran, H.A.; Epps, K.C.; Singh, R.; Psotka, M.; Shah, P.; Cooper, L.B.; Rosner, C.; Raja, A.; Barnett, S.D.; Saulino, P.; deFilippi, C.R.; Gurbel, P.A.; Murphy, C.E.; O’Connor, C.M.	2019	JACC	Low	Low	Low	Low	Low	Low	
The relationship between cardiac reserve and survival in critically ill patients receiving treatment aimed at achieving supranormal oxygen delivery and consumption	Timmins, A.C.; Hayes, M.; Yau, E.; Watson, J.D.; Hinds, C.J.	1992	Postgraduate Medical Journal	Moderate	Low	Low	Low	Low	Low	
Hemodynamic variables and mortality in cardiogenic shock: a retrospective cohort study.	Torgersen, C.; Schmittinger, C.A.; Wagner, S.; Ulmer, H.; Takala, J.; Jakob, S.M.; Dünser, M.W.	2009	Critical Care	Low	Low	Low	Low	Low	Low	
Early worsening heart failure in patients admitted for acute heart failure: time course, hemodynamic predictors and outcome.	Torre-Amione, G.; Milo-Cotter, O.; Kaluski, E.; Perchenet, L.; Kobrin, I.; Frey, A.; Rund, M.M.; Weatherley, B.D.; Cotter, G.	2009	Journal of Cardiac Failure	Low	Moderate	Low	Low	Moderate	Low	
Septic cardiomyopathy: hemodynamic quantification, occurrence and prognostic implications.	Werdan, K.; Oelke, A.; Hettwer, S.; Nuding, S.; Bubel, S.; Hoke, R.; Russ, M.; Lautenschläger, C.; Mueller-Werdan, U.; Ebelt, H.	2011	Clinical Research in Cardiology	Low	Low	Low	Low	Moderate	Low	
Severity of cardiac impairment in the early stage of community-acquired sepsis determines worse prognosis.	Wilhelm, J.; Hettwer, S.; Schuermann, M.; Bagger, S.; Gerhardt, F.; Mundt, S.; Muschik, S.; Zimmermann, J.; Bubel, S.; Amoury, M.; Kloess, T.; Finke, R.; Loppnow, H.; Mueller-Werdan, U.; Ebelt, H.; Werdan, K.	2013	Clinical Research in Cardiology	Low	Low	Low	Low	Low	Low	
Peak exercise cardiac power output; a direct indicator of cardiac function strongly predictive of prognosis in chronic heart failure.	Williams, S.G.; Cooke, G.A.; Wright, D.J.; Parsons, W.J.; Riley, R.L.; Marshall, P.; Tan, L.B.	2001	European Heart Journal	Low	Low	Low	Low	Low	Low	
How do different indicators of cardiac pump function impact upon the long-term prognosis of patients with chronic heart failure?	Williams, S.G.; Jackson, M.; Cooke, G.A.; Barker, D.; Patwala, A.; Wright, D.J.; Albuoaini, K.; Tan, L.B.	2005	American Heart Journal	Low	Low	Low	Low	Low	Low	
Evaluation of Resting Cardiac Power Output as a Prognostic Factor in Patients with Advanced Heart Failure.	Yildiz, O.; Aslan, G.; Demirozu, Z.T.; Yenigun, C.D.; Yazicioglu, N.	2017	The American Journal of Cardiology	Low	Low	Low	Low	Moderate	Low	

**Table 3 healthcare-10-02417-t003:** Detailed Journal Article Demographics, Outcomes and Disease State Definition. Condition Examined.

	Article #	Outcomes	Method of Measuring CP	Timing of When CP Was Measured	Definition of Disease State	Method of Calculating CP	Population (Included and Analyzed)	Age	Sex
TAVR due to AS	22	All-cause mortality, 1 year (post TAVR)	Echocardiography	Prior to TAVR	Symptomatic Severe AS; where severe AS was based on criteria set forth by the American Society of Echocardiography and Society for Cardiovascular Magnetic Resonance.	(MAP × CO)/(451 × BSA)	975	81 ± 8.6	59.4% male, 40.6% female
TAVR due to AS	23	All-cause mortality, 1 year (post TAVR)	Echocardiography	Prior to TAVR	Symptomatic Severe Aortic Stenosis; where severe AS was based on criteria set forth by the American Society of Echocardiography and Society for Cardiovascular Magnetic Resonance.		1055 (907 alive, 148 dead)	80.9 ± 7.9 (81 ± 9 (alive), 80 ± 9 (dead))	58.2% male (58% (alive), 59.5% (dead))
TAVR due to AS	24	All-cause mortality, 1 year (post TAVR)	Echocardiography	Prior to TAVR	Symptomatic Severe Aortic Stenosis; where severe AS was based on criteria set forth by the American Society of Echocardiography and Society for Cardiovascular Magnetic Resonance.	(MAP × CO)/(451 × body surface area (BSA)) (W/m2)	975 (840 alive, 135 dead)	81 ± 9	59.04% male
TAVR due to AS (emergent; in CS)	30	Mortality up to 1 year	PAC	Prior to TAVR	High risk patients with AS, which starting with patient #165 included patients in cardiogenic shock due to AS basing cardiogenic shock on the IABP-SHOCK II trial.	MAP × CI × 0.0022	771 (27 emergent)	80 ± 7	46.8% male
Septic shock	17	28 day mortality	PiCCO (thermodilution)	Cardiac Power data was obtained during a “hemodynamically stable period” which was designated as day 0.	Patients were included in first 12 hours after diagnosing septic shock defined by a SBP < 90 mmHG (or a decrease of >50 mmHg in patients known to be hypertensive), a persisting MAP < 70 mmHg or DBP ≤ 40 mmHg despite adequate fluid resuscitation requiring vasoactive support by norepinephrine >0.05 mu/kg/min during more than one hour. Followed Surviving Sepsis Campaign guidelines.	CI × MAP/451	70	62 ± 16	72.9% (51/70) male
Septic shock	18	30-day mortality (following ED admission)	PiCCO (central line + arterial catheter) or PAC in ICU, transthoracic bioimpedance on wards	At time of admission into the ED	Patients determined to have clinical sepsis which indicated they met at least 2 of the four criteria being Body temperature >38 °C or <36 °C; HR >90; breathing rate >20/min or hyperventilation (PaCO2 < 4.3 kPa); leukocytosis >12,000 or leukopenia <4000 or >10% premature granulocytes. From there patients were classified to stages of sepsis according to the recommendations of the consensus conference of the American Collage of Chest Physicians and the Society of Critical Care Medicine and current sepsis guidelines (sepsis-severe sepsis-septic shock)	CPI = CI*MAP*0.0022; CI = CO/BSA	141	68 (56; 77)	61% male
Septic shock	48	ICU mortality. Correlation with APACHE II and Sepsis scores	PAC	Began once patients were included into the study and continually performed throughout the study.	Patients in medical intensive care/critical care ward who stayed longer than 24 h and having septic Multi-Organ dysfunction syndrome indicating they had an APACHE II score ≥20 and sepsis score ≥12.		39 but 24 obtained hemodynamic measurements	52 (64 for patients with hemodynamic data)	62% male (total), 67% (16/24) in subgroup with hemodynamic data
Post-MI	20	In-hospital, 1 year, 5 year mortality. Secondary outcomes were ischemic complications: recurrent in-hospital ischemia, recurrent nonfatal MI and need for revascularization procedures (PCI or CABG)	Thoracic electrical bioimpedance (HeartLab)	Thoracic electrical bioimpedance was used to assess CP shortly after consent and again on day 3, usually the first 10 s to determine CP.	Patients with ST-elevation myocardial infarct and following diagnosis of acute MI based on criteria provided by the European Society of Cardiology.	(CO × MAP)/451. (CI × MAP)/451	208	63 (53–70)	71.2% male
Post-MI	21	In-hospital mortality (primary). Secondary: complicated in-hospital course (hemodynamically unstable arrhythmia, cardiogenic shock, pulmonary edema, clinically significant recurrent myocardial ischemia, or death)	Thoracic electrical bioimpedance (HeartLab)	Thoracic electrical bioimpedance was used to assess CP shortly after consent and again on day 3, usually the first 10 s to determine CP.	Patients with ST-elevation myocardial infarct and following diagnosis of acute MI based on criteria provided by the European Society of Cardiology.		232 (67 w/diabetes, 165 w/o)	62.8 (10.7)	69.8% male (73.9% in diabetic, 59.7% nondiabetic, *p* = 0.032)
Critical Cardiac Disease (CCU)	3	In-hospital mortality	PAC (average of 3–5 readings)	Used baseline data for patients admitted to the CCU.	Patients with primary cardiac diagnosis who were undergoing PAC, most common primary diagnosis were myocardial infarct, cardiomyopathy, coronary atherosclerosis and unstable angina.	MAP × CO/451	349	64 ± 14	59% male (64% for CPO >0.53, 34% for CPO ≤0.53, *p* < 0.001
Critical Cardiac Disease (CICU)	35	In-hospital mortality	Echocardiography	Patients had transthoracic echocardiogram within the first day of CICU admission.	Patients were included that were admitted to the CICU and met any stage of the Society for Cardiovascular Angiography and Intervention shock stages.	(CO × MAP)/451. MAP = (SBP + (2 × DBP))/3	5453	69.3 (58.2–79.0)	63.3% male. 2.001 (36.7) female
Heart failure	15	Patient followed up for minimum of 2 years or until death.	CO2 rebreathing	Outpatient cardiopulmonary treadmill exercise testing	Mild-moderate chronic stable heart failure patients	CPO = (CO*MAP)*0.00222. MAP = (SBP + 2*DBP)/3	219	56.14 ± 13.13	76% male
Heart failure	16	All-cause mortality, heart transplant, or VAD placement (median follow-up 3.3 years)	PAC	Outpatient pulmonary artery catheterization	Advanced chronic heart failure for greater than six months	MAP × CO × 0.0022	495	54 ± 11	75.8% male
Heart failure	31	Mortality up to 1 year	Echocardiography	Hemodynamic measurements were taken preoperatively and at 6 and 12 h postoperatively.	Patients with HF that led to cardiogenic shock which was defined as unable to meet systemic circulatory demands without supportive measures, except for correction of volume or vascular resistance.	CO × MAP/451	50 (33 surgical, 17 non-surgical	54.5 (58.1 surgical, 47.5 non-surgical)	70% male
Heart failure	32	All-cause mortality (up to 5 years)	PAC	Hemodynamic measurements were taken pre-left ventricular assist device implantation	Patients with advanced heart failure who are receiving left ventricular assist device	(MAP × CO)/451	18733		
Heart failure	33	Survival at 90 days post MCS	PAC and/or echo	Hemodynamics measurements were taken when patients were first admitted and prior to any inotropic drug therapy.	Patients with New York Heart Association classes II and IV and American College of Cardiology stages C and D.	(MAP × CO) × 0.0022]/BSA. MAP = ((2 × diastolic) + systolic/3	28	53 ± 2.5	85.7% male (24/28)
Heart failure	38	A composite endpoint was used as the outcome, which included cardiac-related death, hospitalization for worsening HF, cardiac transplantation and left ventricular assist device (LVAD) implantation. Mean follow up was 460 ± 332 days	Thoracic bioimpedance	Hemodynamic measurements were obtained upon initial evaluation	Patients initially evaluated for heart failure presentation.	peak CPO = peak (MAP × CO)/451	639	48.2 ± 14	62% male
Heart failure	40	All-cause mortality, VAD implantation and heart transplantation was the primary endpoint. Combination of the former with hospitalization for worsening HF was designated as the secondary endpoint. Follow up 34.5 (10.9–57.6) months	Echocardiography	Initial hemodynamic workup was performed upon initial presentation at university.	Chronic systolic heart failure with LVEF ≤40%	CPO = k*CO*MBP. K = 2.22*10^−3^. MBP = diastolic BP + 1/3 (systolic BP–diastolic BP). CPOM = (CPO*100)/LVM	159	62 (56–70)	81.8% male
Heart failure	42	Death, heart transplant, or LVAD implantation. Follow-up averaged 404 ± 179 days (median, 366 days)	Bioimpedance	Initial hemodynamic workup was performed upon initial presentation at medical center.	Moderate to advanced chronic heart failure	CPO = (CPO*MAP)/451. MAP = DBP + (SBP − DBP)/3	127	53 ± 14	66% male (40% with event, 69% without, *p* = 0.0388)
Heart failure	43	1 year mortality or major events (serious ventricular arrhythmia, acute pulmonary edema, or hospitalization for HF)	PAC	Initial hemodynamic workup was performed upon initial presentation.	Stable chronic systolic heart failure with New York Heart Association class II or III.	(MAP-RAP)*CO*0.00222. MAP = DBP + (SBP − DBP)/3	50	54.5 ± 1.6	64% male
Heart failure	47	WHF at 7/30 days, 6 month mortality. WHF: new pulmonary edema or cardiogenic shock, no resolution of symptoms and signs of HF in first 24 h despite therapy, or worsening symptoms and signs of HF despite therapy along with an increase in treatment for HF (IV, vent, MCS)	PAC	Baseline hemodynamic values were obtained at initial presentation.	Acute heart failure requiring hemodynamic monitoring	CPO = MAP*CO*0.0022	120 (42 WHF in 7 days, 50 WHF in 30 days, 17 deaths)	64.2 ± 13.6	86% male
Heart failure	49	Patients followed for 6 years or until death (median follow up 8.6 years).	CO_2_ rebreathing	Baseline hemodynamic values were obtained at initial presentation.	Mild to moderate chronic systolic heart failure.	CPO = (Total O_2_ consumption/(O_2_)pulmonary veins-(O_2_) pulmonary arteries))*MAP	219	56.14 ± 13.13	76% male
Heart failure	50	All-cause mortality, VAD placement, or heart transplant	Right and left cardiac catheterization	Hemodynamic measurements were obtained upon initial evaluation	Advanced chronic heart failure severe enough to warrant cardiac tertiary care institution.	CPO = (MAP*CO)/(0.00222), CPI = (CI*MAP/0.00222). MAP = ((SBP-DBP)/3) + DBP	161	58.7 (11.2)	73.9% male
Cardiogenic shock	6	In-hospital mortality	PAC/cardiac catheterization	Hemodynamic measurements were made within 6 h pre shock and 12 h post-shock.	Predominant left ventricular failure causing cardiogenic shock determined form the SHOCK trial registry and eliminating other categories of cardiogenic shock such as cardiac tamponade, severe valvular heart disease, dilated cardiomyopathy, ventricular septal rupture and more which was determined based on clinical grounds.	MAP × CO/451; MAP = ((SBP − DBP)/3) + DBP	541	67.8 ± 12.4	65.3% male
Cardiogenic shock	11	30 day (post-discharge) mortality	PAC (82%) and echo	Cardiac Power data was obtained once shock definition was met and repeated at 24 h after cardiogenic shock was diagnosed.	Shock was determined based on the following definition: as SBP < 90 mmHg for >30 min (or use of inotropes/vasopressors to maintain SBP), evidence of end-organ hypoperfusion and lactate level >2 mmol. Afterwards cardiogenic shock was defined with Fick cardiac index <1.8 L/min/m^2^ without inotropes/vasopressors (or <2.2 L/min/m^2^ with inotropes/vasopressors), PCWP >15 mmHg, CPO < 0.6 W, pulmonary arterial pulsatility index <1.0.		204	61 ± 13	70% male (for AMI 70.7%, for ADHF68.8%)
Cardiogenic shock	12	30-day mortality	PAC	Hemodynamic measurements were collected at 0,6,12,24,48,72 and 96 h after detection of shock.	Cardiogenic shock required SBP <90 mmHg in absence of hypovolemia and after adequate fluid challenge for 30 min or need for vasopressor therapy to maintain SBP >90 mmHg, symptoms and/or signs of systemic and/or pulmonary congestion and symptoms and/or signs of hypoperfusion (Altered mental state, confusion, cold periphery, oliguria <0.5 mL/kg/h for the previous 6 h, blood lactate >2 mmol/L) all of which was determined to be due to cardiac etiology.	(MAP*CI)/451	219 (82 w/PAC, 137 w/o)	65(12) w/PAC, 68(11) w/o, *p* = 0.09	74.0% male overall (78% w/PAC, 72% w/o)
Cardiogenic shock	13	Event free survival for 28 days. Adverse events defined as death or emergent MCS escalation, favorable outcomes include survival to discharge, successful bridge to durable LVAD, or transplant	PAC	Hemodynamic measurements performed prior to intra-aortic balloon pump placement.	Patients undergoing intra-aortic balloon pump for cardiogenic shock.	(CI × MAP)/451. RVCPI = (CI × mPAP)/451	74	54.8 ± 14.1	66% male
Cardiogenic shock	14	30 day mortality	PAC (71%), CVP (29%); thermodilution	Baseline hemodynamic values were obtained at initial presentation.	Patients were admitted initially for acute myocardial infarct but presented with cardiogenic shock defined by sustained hypotension (systolic blood pressure < 90 mmHg) induced by heart failure together with the clinical signs of hypoperfusion (cold extremities, oliguria, or altered mental state), not responsive to fluid resuscitation.	(MAP × CI)/451	68	60 ± 14	69% male
Cardiogenic shock	25	Increase in CO within 15 h of IABP placement. Also looks at 60/90 day survival	PAC	All hemodynamic measurements were obtained at baseline evaluations.	Cardiogenic shock was defined as persistent hypotension with SBP ≤90 mmHg or MAP 30 mmHg lower than baseline and reduction in CI ≤ 1.8 L/min/m^2^ without inotropic support or <2.2 L/min/m^2^ with inotropic support and adequate filling pressure with left ventricular end diastolic pressure ≥18 mmHg or Right ventricular end diastolic pressure >10 mmHg.	(MAP × CO)/451, (MAP × CI/451)	76	55.6 ± 13.2	80.2% male
Cardiogenic shock	26	In-hospital mortality	PAC (92% of patients, hemodynamic data only recorded from these patients)	Hemodynamic measurements were taken both pre-procedure and post-procedure.	Cardiogenic shock was defined as persistent hypotension with SBP ≤90 mmHg or inotropes/vasopressors to maintain SBP >90 mmHg, signs of end organ hypoperfusion (cool extremities, oliguria or anuria or elevated lactate levels and hemodynamic criteria represented by CI <2.2 L/min/m^2^ or CPO <0.6 W.	(MAPxCO)/451	171	63.4 ± 12.4. Non-survivors/survivors: 63.4 ± 12.4/61.4 ± 12.6, *p* < 0.01	77.2% male (68.8%/80.5% non-survivors/survivors)
Cardiogenic shock	34	30 day mortality	PAC	Baseline hemodynamics were obtained at 3.3 h after cardiogenic shock and follow up values were measured with median time 10.6 h in the medical stabilization group and 12.5 h in the early vascularization group.	Cardiogenic shock was defined as SBP <90 mmHg for 30 min or supportive measures such as vasopressors or intra-aortic balloon counter pulsation required to maintain a blood pressure of ≥90 mmHg with evidence of decreased organ perfusion (urine output of ≤30 mL/h or cool and diaphoretic extremities and a HR of ≥60 beats/min). Hemodynamic criteria for cardiogenic shock were pulmonary capillary wedge pressure ≥18 mmHg and CI of ≤2.2 L/min/m^2^.	(MAP × CI)/451. MAP = (systolic BP − diastolic BP)/3 + diastolic BP	278	66 ± 11	67% male
Cardiogenic shock	37	Mortality at 28 days, 1 year	PAC/PiCCO (thermodilution)	Hemodynamic measurements were made on initial workup for patients in the intensive care unit.	Cardiogenic shock was defined as SBP <90 mmHg in the absence of hypovolemia or vasopressors, a reduction in CI <1.9 L/min/m^2^ and or an elevation of pulmonary capillary wedge pressure >19 mmHg all due to a cardiac etiology.		96 (60 survived to 28 days, 37 to 1 year)	Survivors: 66 (20; 90), non-survivors: 70 (43; 86)	72.9% male. 41/60 male (survivors), 29/36 (non-survivors)
Cardiogenic shock	39	In-hospital mortality. Death, new MI, urgent PCI, stroke, sepsis, major hemorrhagic complications according to TIMI criteria. Follow up 28 days and 1.9 ± 0.9 years post discharge	PAC/PiCCO (thermodilution) and echo	All hemodynamic measurements were obtained at baseline evaluation.	Cardiogenic shock due to myocardial infarction which was defined by SBP < 90 mmHg for 30 min or MAP of less than 60 mmHg for 30 min or vasopressor use with evidence of decreased organ perfusion. The manifestations of hypoperfusion could include, but are not limited to lactic acidosis, oliguria, or an acute alteration in mental status. Hemodynamic criteria were pulmonary capillary wedge pressure of more than 15 mmHg and a cardiac index of less than 2.2 L/min/m^2^ in patients not treated with vasopressor/inotropes.		85	64(15). 62(12) for S, 66(13) for NS, *p* > 0.05	74% male overall. 31% female in S, 19% in NS, *p* > 0.05
Cardiogenic shock	41	Mortality ICU, 28 d	PAC/PiCCO (thermodilution; 69/71), but some echo	Hemodynamic variables were collected during the first 24 h after ICU admission.	Cardiogenic shock was defined as SBP < 90 mmHg or MAP< 60 mmHg for 30 min with or without therapy; the need for continuous infusion of inotropic drug; a CI less than 1.8 L/min/m^2^ without inotropes or less than 2.2 L/min/m^2^ with inotropes; and a pulmonary artery occlusion pressure greater than 18 mmHg. Classical clinical criteria of shock indicating low organ perfusion were not included in the criteria.		71	65 ± 14	52 (73%) male
Cardiogenic shock	44	1-year mortality	PAC	Measurements were done at baseline resting state in patients as well as after optimalisation of preload and usage of dobutamine.	Shock was diagnosed according to Cohn’s criteria which required them to meet two of the following criteria including oliguria with urine output <20 mL/h, cool moist skin, auscultatory SBP <90 mmHg, obtunded mental state, metabolic acidosis all of which should be due to cardiac etiology.	(MAP − RAP) * CO * 2.2167 * 10^−3^	28	59 (36–73)	75% male
Cardiogenic shock	46	28 day mortality after ICU admission. Length of ICU/hospital stay, patient outcome at ICU discharge.	PAC (77.2% of subjects; CPI measurements only done for these patients with PAC; values from first 24 h in ICU)	Hemodynamic measurements were obtained within the first 24 h from intensive care unit admission.	Cardiogenic shock was defined as simultaneous presence of all the following criteria immediately before or during the first 24 h after intensive care unit admission: arterial hypotension (SBP <90 mmHg or MAP below 70 mmHg for 30 min or longer with or without therapy), a cardiac index below 2 L/min/m^2^ and a pulmonary artery occlusion pressure above 18 mmHg in patients with a pulmonary artery catheter or an acute decrease of the left ventricular ejection fraction below 40% in patients without a pulmonary artery catheter; need for continuous infusion of inotropic drugs.	CPI = MAP*CI/451	119 (23 ICU mortality, 35 28 day mortality)	67 ± 14	59.7% male
Other- Extracorporeal Circulation	27	Survival up to 7 days post-procedure	Indocyanine green method	Hemodynamic variables were obtained within the first 26 h after surgery.	Patients who underwent extracorporeal circulation and a cardiac procedure who were admitted directly to a cardiothoracic surgical intensive care unit from the operating room.	CO(MAP-CVP)	181	2–72 (mean 46)	59.1% male
Other- Ischemic Cardiomyopathy	28	Mortality (median follow-up of 29 months (inter-quartile range 16–72 months))	Echocardiography	Hemodynamic measurements were done with the use of dobutamine during initial workup.	Ischemic cardiomyopathy with left ventricle ejection fraction ≤45% which is not due to other cardiac pathologies.	MBP (diastolic BP + 1/3 (systolic BP − diastolic BP)) × CO × 2.22 × 10^−3^. CPO/LVM = MAP*CO*100*2.22*10^−3^/LVM	111	68 ± 10	85.6% male
Other- Chronic Kidney Disease	29	All-cause mortality, 12 ± 1 months after the monitored session	Bioimpedance (whole body)	Patient hemodynamic measurements were obtained during a single hemodialysis session.	Patients were determined to have CKD if they had undergone chronic hemodialysis for at least 3 months.	MAP × CI/451	144	67.3 ± 12.1	56.3% male
Other- Heart Transplant	36	Severe PGD (urgent MCS or death within 30 days of OHT)	PAC	Hemodynamic measurements upon return to the intensive care units were obtained at time 0 and 6 h.	Patients who underwent heart transplantation.	(CI × (MAP-CVP))/451. CPI adjusted for vasoactive-inotrope score (VIS) = (CPI/√VIS + 1) × 100	140	No PGD 48 (34–56), severe PGD 55 (50–60)	69.3% male overall (no severe PGD 69%, severe PGD 73%)
Other- Non-cardiac related illness	45	In-hospital mortality	PAC	Data was obtained/analyzed on admission known as t_0_, after optimal volume resuscitation known as t_1_ and at maximal resuscitation known as t_max_.	Patient indicated to have non-cardiac critical illness included patients without cardiac etiology that had septic shock according to the Bone criteria and having adult respiratory distress syndrome which was defined as the following: compatible underlying clinical disorder, inspired oxygen fraction of greater than 0.4 to maintain a PaO2 of greater than 9 kPa, X-ray evidence of diffuse bilateral pulmonary infiltrates and pulmonary artery occlusion pressure < 18 mmHg.	CPO = (MAP-RAP)*CO*2.22*10^−3^; CPI= CPO/BSA	32 (15 survivors)	Survivors: 54 (19–80), non-survivors: 67 (27–80)	71.9% male (23/32)

Abbreviations: PAC = Pulmonary Artery Catheter, MAP = Mean Arterial Pressure, BSA= Body Surface Area, CVP = Central Venous Pressure, Echo = Echocardiography, PiCCO = Pulse index Contour Continuous Cardiac Output, SBP = Systolic Blood Pressure, LVEF = Left Ventricular Ejection Fraction, AS = Aortic Stenosis.

## Figures and Tables

**Figure 1 healthcare-10-02417-f001:**
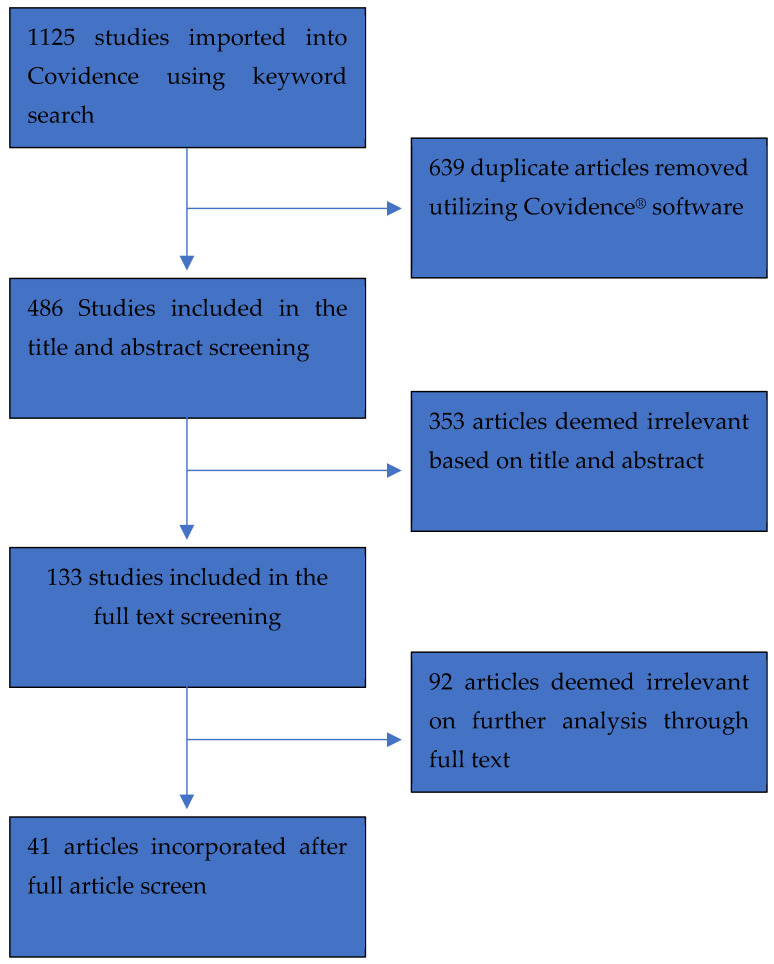
Flow Diagram for Study Screening and Data Collection.

**Table 1 healthcare-10-02417-t001:** Categorical organization of articles in literature review based on disease process.

Disease State Category	Articles Included
Transcatheter Aortic Valve Replacement	[19,20,21,22]
Septic Shock	[17,18,23]
Post-Myocardial Infarct	[24,25]
Critical Cardiac Disease	[3,26]
Heart Failure	[15,16,27,28,29,30,31,32,33,34,35,36]
Cardiogenic Shock	[6,11,12,13,14,37,38,39,40,41,42,43,44]
Other	[45,46,47,48,49]

## Data Availability

Not applicable.

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
