# Peer review of "Utility of Cardiac Power Hemodynamic Measurements in the Evaluation and Risk Stratification of Cardiovascular Conditions"

_healthcare, 2022, doi:10.3390/healthcare10122417_

Round 1

Reviewer 1 Report

The submitted manuscript in form of an educational review with title: "Utility of Cardiac Power Hemodynamic Measurements in the Evaluation and Risk Stratification of Cardiovascular Conditions" , despite its retrospective nature of the selected observational studies coped with their respective bias and heterogeneity in the patient selection for evaluation cardiac power, was clearly presented. 

To define the relevance of CP-Hemodynamics in the selected patient collective should be further assessed. The presented table-overview of the relevant articles was assessed logically according to their utility in a clinical setting. 

All the potential flaws of the review were analysed and presented properly. 

Author Response

Please see the attachment below and thank you.

Reviewer 2 Report

Utility of Cardiac Power Hemodynamic Measurements in the Evaluation and Risk Stratification of Cardiovascular Conditions

Farshadmand and co-authors present a review of the literature examining the utility of cardiac power measurements in predicting outcomes in various cardiac conditions
Medical databases were searched for articles containing the term "cardiac power" in either the title, abstract, or keywords
Inclusion criteria:
- English language
- Full text articles
- Evaluation of resting CP measurementin relation to mortality
Exclusion criteria:
- Animal studies
- Case reports
- Abstracts
- Review articles
- Commentaries on published studies
- Studies involving pediatric populations (18 years and younger)
- Studies that evaluated a surrogate or proxy for CP
- Studies that did not directly associate CP with mortality
- Studies that used CP as an end point

41 relevant studies were identified and screened for bias using QUIPS tool. Results were presented for 6 different conditions: cardiogenic shock, septic shock, TAVR, heart failure, critical cardiac illness, and other conditions. The conclusion of the review is that cardiac power measurements are a good predictor of mortality in cardiogenic shock, chronic heart failure, and sepsis.

General comments:
Thank you for the opportunity to review the manuscript. It is a nice collection of relevant literature on this topic, but it lacks some new information. The authors should try to compensate for the lack of novelty by presenting the data in a more structured way that is easier to understand.
As with all interventions and prognostic tools, it is important to know when the test was performed. This is not apparent from the manuscript. Could the different timing of cardiac power measurement explain the different results of the studies? Specifically, in septic shock, was cardiac power measured on admission or later in the course of the illness? Similarly, for other disease-specific categories.
A flowchart for selecting relevant literature might be useful to facilitate description of the search and selection process
Providing details on how cardiac power measurements were performed in the evaluated studies could facilitate comparison.
Major comments:
Page 2-3, lines 88-95, Methods, Study screening and data collection: could you provide a more detailed description of the different disease categories. Some are obvious (TAVR), others less so (critical cardiac disease). Also, there are different definitions of cardiogenic shock, and the definition of sepsis has changed in recent years. The criteria for assigning a particular study to a disease-specific category should be outlined (possibly in the body of the manuscript or in the appendix).
Page 4, lines 168-201, Heart failure: could you better describe the heart failure population. Are they patients with stable or decompensated heart failure? Were the heart power measurements performed for the inpatient or outpatient population?
Minor comment:
A table with relevant data for each evaluated manuscript (number of patients, definition of evaluated population, how and when cardiac power was measured, outcome ....) would be interesting and useful.

Reviewer 3 Report

The authors of the current manuscript are making a review of published articles, focused on prognostic hemodynamic parameters, trying to determine the prognostic importance of some cardiac power measurements such as cardiac power output and cardiac power index in various cardiac pathologies.

The manuscript is generally well-structured and comprehensive – the topic should be of interest to the readers, who work clinical cardiovascular Medicine. I have the following recommendations to the authors about some modifications/additions to their paper:

Line 32-33: I do not agree with this statement - “difficulty in determining the severity of various cardiac conditions”. The large prevalence of cardiovascular diseases and the high mortality and morbidity rates caused by it are mostly due to lack of control of the cardiovascular risk factors (insufficient primary prevention and promotion of health), delayed initial diagnosis, insufficient adherence to the prescribed therapies and other (mostly subjective factors), but not due to lack of appropriate quantification of the severity of already diagnosed conditions (or lack of effective therapies).  This part of the text should be modified.

 Please, spell the words of the abbreviation “CP” the first time it appears in the text (in “Introduction”)

Round 2

Reviewer 2 Report

I don't have any further comments or suggestions. All my concerns were adequately addressed.